# A Data Augmentation Method for Deep Learning Based on Multi-Degree of Freedom (DOF) Automatic Image Acquisition

**Liangliang Chen, Ning Yan, Hongmai Yang, Linlin Zhu, Zongwei Zheng, Xudong Yang and Xiaodong Zhang \***

State Key Laboratory of Precision Measuring Technology & Instrumuments, Centre of MicroNano Manufacturing Technology, Tianjin University, Tianjin 300072, China; liangliangchen@tju.edu.cn (L.C.); yanning@tju.edu.cn (N.Y.); 3014202025@tju.edu.cn (H.Y.); l_linzhu@tju.edu.cn (L.Z.); zongweizheng@tju.edu.cn (Z.Z.); 3014202026@tju.edu.cn (X.Y.)

**\*** Correspondence: zhangxd@tju.edu.cn; Tel.: +86-138-2103-3646

**Abstract:** Deep learning technology is outstanding in visual inspection. However, in actual industrial production, the use of deep learning technology for visual inspection requires a large number of training data with different acquisition scenarios. At present, the acquisition of such datasets is very time-consuming and labor-intensive, which limits the further development of deep learning in industrial production. To solve the problem of image data acquisition difficulty in industrial production with deep learning, this paper proposes a data augmentation method for deep learning based on multi-degree of freedom (DOF) automatic image acquisition and designs a multi-DOF automatic image acquisition system for deep learning. By designing random acquisition angles and random illumination conditions, different acquisition scenes in actual production are simulated. By optimizing the image acquisition path, a large number of accurate data can be obtained in a short time. In order to verify the performance of the dataset collected by the system, the fabric is selected as the research object after the system is built, and the dataset comparison experiment is carried out. The dataset comparison experiment confirms that the dataset obtained by the system is rich and close to the real application environment, which solves the problem of dataset insufficient in the application process of deep learning to a certain extent.

**Keywords:** deep learning; image acquisition; camera position; path optimization

## 1. Introduction

As one of the branches of machine learning, deep learning forms more abstract and high-level features by combining simple features at the bottom, so as to extract and represent complex features of input data [1]. With the development of deep learning in recent years, it has shown superior performance in many fields. Nowadays, deep learning has become the focus of many scholars' attention in the field of image recognition and image analysis [2–7]. A dataset is the base of deep learning and the final effect of a deep learning model largely depends on the comprehensiveness and authenticity of training data.

Therefore, enterprises and institutions engaged in deep learning always spend a lot of time and energy collecting datasets. For example, Microsoft has made the dataset named MS-COCO with 328,000 images, Google has made the dataset named Open Image V4 with 1,900,000 images. Expensive cost investment makes deep learning training datasets valuable. To solve the difficult problem of dataset acquisition, scholars have put forward some solutions. One method is to install camera equipment in the detection environment, and multi-attitude datasets can be obtained with

the movement of the structure to be tested. Liu et al. [8] collect body posture by installing an indoor camera to acquire the different postures of people in the process of daily activities. Berriel et al. [9,10] obtained a road dataset by vehicle cameras. However, these ways are hardly applied in specific industrial visual inspection. For example, it is necessary to classify and identify different kinds of parts for convenient access in the field of automobile parts production. Meanwhile, in the production process, scratches, bumps, and other defects need to be detected in time to guide the processing and production [11–13]. However, the collection process is time-consuming and labor-intensive. In the field of textile weaving, there is a wide variety of fabrics produced and different categories have a high degree of similarity. In addition to the rapid update of fabric categories, the realization of rapid and accurate identification of fabric types has instructive significance for production and sales. Besides, various defects will inevitably occur in the production process, and the detection of different fabric defects is also very important for the textile industry [14–17]. However, there is no perfect solution for the fabric category and fabric defect image collection. In the above application fields, the image acquisition environment is complex, which needs to be shot in multi-pose and multi-illumination directions to obtain comprehensive datasets. It ensures that the final deep learning model adapts to various complex situations. However, the existing automatic data's acquisition methods cannot meet these kinds of demands. Other data acquisition methods focused on extending the original data [18,19]. Two typical methods are traditional digital image processing technology and neural networks. Traditional digital image processing technology increases the number of images by rotating, translating, clipping, flipping, converting color space, adding noise, mixing images, random erasing, and random combinations of different operations. Alexander Jung [20] has published common flip, rotation, cropping, deformation, zooming, and other geometric transformations and color transformations and other single-sample enhancement techniques on the Internet. Zhong et al. [21] proposed a data augmentation method of random erasure. During the training process, the training pictures are occluded to improve the robustness of the model and reduce the risk of overfitting; Inoue et al. [22] used the SamplePairing technique to synthesize a new sample by randomly selecting two images from the training data (the average value of the two images per pixel of the image). Zhang et al. [23] proposed the data augmentation method of Mixcut. First, a crop box was randomly generated to crop the corresponding position of the A picture, and then the region of interest (ROI) of the corresponding position of the B picture was placed in the cropped area of the A picture to form a new sample. The method based on a neural network is to generate new image samples based on the original data through the neural network models such as variational autoencoder (VAE) [24] and generative adversarial networks (GAN) [25]. Zhu et al. [26] proposed a data augmentation method using GAN. The results showed that the use of GAN-based data augmentation techniques can increase the accuracy of emotion classification by 5% to 10%. However, though a large number of data can be generated by data augmentation, augmented data are not always representative enough due to the complexity of the shooting environment, conditions, and methods, such as the illumination diversity of the data in practical application, resulting in that the final model has low accuracy in real data detection. In summary, in the field of deep learning, image data acquisition is still a key technical difficulty and has not been solved well.

In view of the insufficient datasets in the practical industrial application of deep learning, we propose a data augmentation method for deep learning based on multi-DOF automatic image acquisition and build a multi-DOF automatic image acquisition system for deep learning to solve the problem of difficult image acquisition. The system can comprehensively simulate the actual image acquisition situation and quickly establish a large number of accurate and real training data. The simulation of the real acquisition situation is carried out through changing the imaging's main factors which are shooting angles and illumination conditions. The system mainly involves two key technologies: The first is to ensure the camera can shoot the object through the calculation of the camera's spatial pose accurately. The other is to optimize the spatial random acquisition path to acquire image data rapidly. In order to verify the authenticity and richness of the dataset obtained by the system, we took fabrics as the research object, then used the fabric data collected by the system and the

fabric data collected under ideal conditions to conduct fabric classification experiments. The results show that the model trained by the system collected dataset can recognize fabrics in different scenes, and the recognition accuracy rate was more than 91%. It shows that the images obtained by the system were more abundant including all kinds of actual acquisition situations.

The structure of the paper is as follows: Section 2 proposes a method of the multi-DOF image automatic acquisition and designs the system architecture. Furthermore, the process of the acquisition system is also introduced. Section 3 includes two key technologies: camera pose calculation and random acquisition path optimization. By comparing two methods of calculating camera pose, the formula of the camera pose is determined. In Section 4, dataset acquisition and dataset comparison experiments were carried out to verify the representativeness of images collected by this system. Section 5 concludes the paper.

## 2. The Method of Multi-DOF Image Automatic Acquisition

In practical industrial applications, due to the complexity of the image acquisition environment, the angle of the acquired image is variable, and the illumination is different. In order to simulate the image acquisition situation comprehensively in the industrial application, we designed a multi-DOF automatic image acquisition system. This system can collect data under different shooting angles and different illumination conditions. Figure 1 shows the general idea of the system design. The system is composed of a motion control module and an illumination control module. The lighting control module takes random lighting in different directions to construct different lighting conditions. The motion control module controls the random motion of multi-degree of freedom motion axis to realize different angles in the spatial range. Considering the relative position between the industrial camera and object and the acquisition time-consuming problem, it is necessary to calculate the camera's spatial pose and optimize the whole random acquisition path when controlling the random motion of the multi-DOF motion axis.

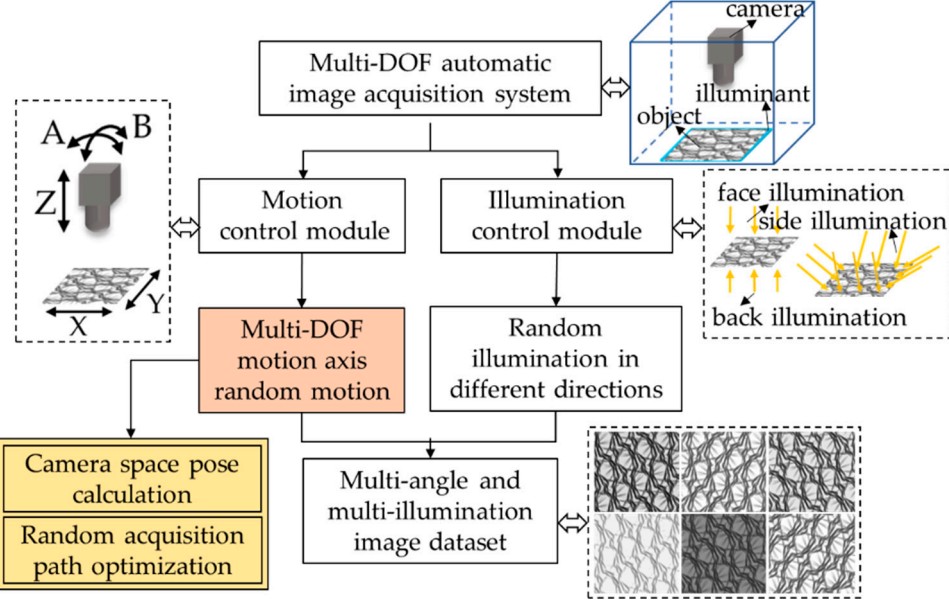

**Figure 1.** The design idea of multi-DOF automatic image acquisition system.

### 2.1. The Design of System Architecture

Figure 2 is the structure diagram of the designed a multi-DOF automatic image acquisition system, which is mainly composed of an industrial camera, industrial lens, 5-DOF motion axis ($x$, $y$, $z$, $a$, $b$), flat illuminant source, circular illuminant source, computer, etc. Specifically, the $x$-axis and $y$-axis drive the object to move randomly in the plane of $x$-, $y$-, and $z$-axis, and the $a$-axis, and $b$-axis drive

the camera to move randomly in space. The flat illuminant source is placed on the inferior and inner wall of the system. The annular illuminant source is mounted on the camera's lens for front lighting. In the actual image acquisition process, the external conditions, such as the camera shooting angle, field of view range, illumination conditions. and other external conditions, will vary. The position and posture of the camera are variable, so the actual image acquisition process can be simulated, and the corresponding design is a camera with two rotation degrees of freedom. Moreover, the height of the camera is adjustable in the vertical direction to realize the image acquisition of a different field of view. For the object to be photographed, the two degrees of freedom displacement platform was designed, which can move the object randomly in the horizontal direction to realize the shooting of different areas of the object. In order to simulate the diversity of lighting modes, the system was set upside, front, and back lighting. The front lighting can highlight the surface morphology of the object; the lateral lighting is to simulate the illumination in different directions; for some complex objects, backlighting is used to highlight the contour information of the object.

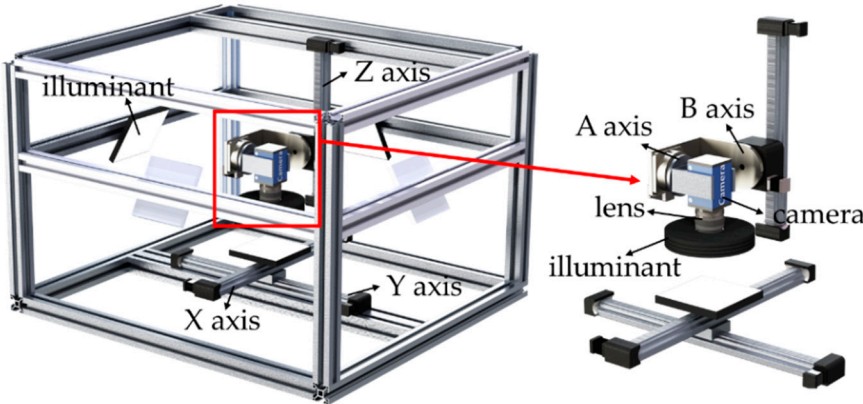

**Figure 2.** Structure diagram of the multi-DOF automatic image acquisition system.

The system needs to shoot the object from multiple angles and illuminant sources (as shown in Figure 3), the camera rotates with two degrees of freedom in the A and B directions and moves vertically in the Z direction; the object moves in the X and Y directions in the plane X–Y, so as to realize the photographing of the object in different positions. Different directions of the front, side, and back are used to shoot the object under different lighting conditions.

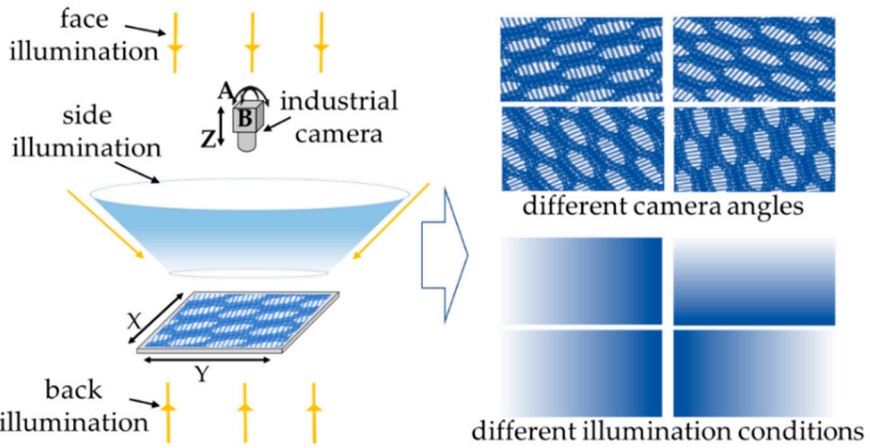

**Figure 3.** Multi-angle and multi-light source shooting schematic diagram.

## 2.2. The Process of System Acquisition

The specific acquisition process is shown in Figure 4: The object is placed on the surface of the backlight source, and then the computer sends out the command to control the multi-axis motion. The *x*-axis and *y*-axis drive the object to move randomly in the plane after receiving the command. Meanwhile, *z*-axis, *a*-axis, and *b*-axis drive the industrial camera to move up and down, tilt and pitch in space after receiving the command. When the object moves randomly to a certain position, the industrial camera moving in space can track the position of the object (the center point of the camera's field of view coincides with the center of the object), the computer sends out the command to control the random illumination. At this time, the flat light sources on the inner wall of the system's cavity, the circular light source carried by the camera and the backlight light source mounted on the object are turned on or off randomly. Consequently, the camera can grab the images under different lighting conditions. Different objects move along random paths in a period, the camera collects images under different angles and illumination conditions, the whole acquisition process simulates all kinds of practical shooting conditions and obtains rich and diverse data.

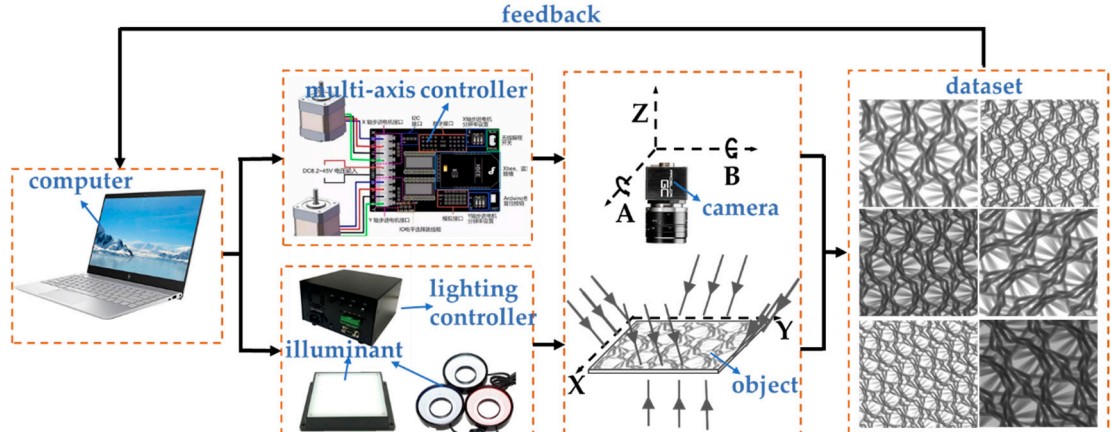

**Figure 4.** Image acquisition flow diagram.

## 3. The key Technology of the Multi-DOF Automatic Image Acquisition System

The key metrics of this multi-DOF automatic image acquisition system are the high quality of dataset (rich and diverse data) and short acquisition time. Therefore, the system mainly involves two key technologies: the coordination of the camera angle and the moving position of the object as well as the optimization of the random motion path.

### 3.1. The Calculation of Camera Position

In the calculation of the camera pose, this paper adopted two methods: mathematical modeling and data fitting, and it compared the two methods. The introduction of these two methods is below.

### 3.1.1. Camera Position Calculating Method by Mathematical Modeling

In the process of object movement, it is necessary to ensure that the image is in the camera field of view, that is, the camera needs to be aligned with the object through the rotating axis. In order to meet this constraint, we established a mathematical model to study the multi-axis geometric relationship as shown in Figure 5. The coordinate system was established with the center of the initial moment of the object being photographed as the coordinate origin, the coordinates of the object to be photographed were $O$ (0, 0, 0) at this time. At the initial moment, the camera direction was vertically downward. The camera installation was set as $h_0$, and the camera coordinates were $C$ (0, 0, $h_0$) as shown in Figure 5a. Suppose that the camera is in focus by moving up $z_0$, and the camera coordinates are $C'$ (0, 0, $h_0 + z_0$)

as shown in Figure 5b. Control the *x*- and *y*-axes to move $\Delta x$ and $\Delta y$, respectively. At the same time, for keeping the distance between the camera and the object unchanged, it is necessary to control the *z*-axis motion $\Delta z$ to ensure that the distance between the camera and the object is $h_0 + z_0$. At this time, the center coordinates and camera coordinates of the photographed object become $O'$ $(\Delta x, \Delta y, 0)$ and $C'$ $(0, 0, h_0 + z_0 - \Delta z)$, as shown in Figure 5c. To guarantee that the camera can shoot the object, rotation axis A and B are controlled to rotate the corresponding angle to ensure that the camera optical axis points to the object to be photographed, as shown in Figure 5d. The specific process, firstly, is the *a*-axis controls the camera rotation $\alpha$, and then the *b*-axis controls the camera rotation $\beta$, given that,

$$
\begin{aligned}
\tan(\alpha) &= \frac{\Delta x}{h_0 + z_0 - \Delta z} \\
\tan(\beta) &= \frac{\Delta y}{\sqrt{\Delta x^2 + (h_0 + z_0 - \Delta z)^2}} \\
\Delta z &= h_0 + z_0 - \sqrt{(h_0 + z_0)^2 - \Delta x^2 - \Delta y^2}
\end{aligned}
\tag{1}
$$

Based on Equation (1), the rotation angle of the rotating shaft can be calculated.

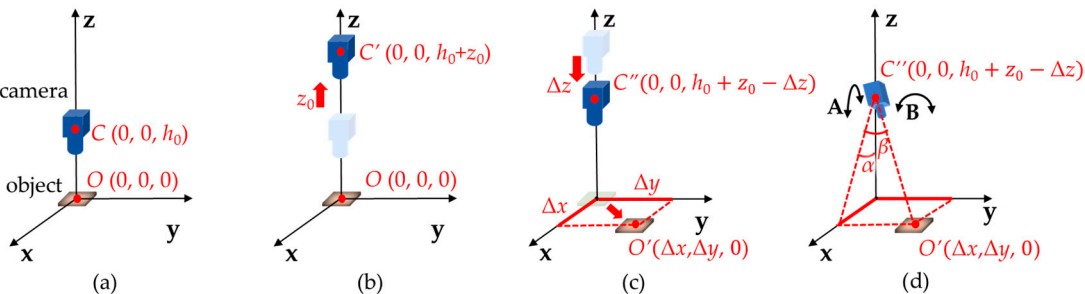

**Figure 5.** The model of the relationship between camera angle and the motion of the object to be photographed: (**a**) the position of the camera installed at the initial moment and the position of the object to be photographed; (**b**) the position of the camera in the focusing state; (**c**) the position after the *x*-, *y*-, *z*-axes movement; (**d**) the rotation axis of the camera controls the camera optical axis to aim at the shooting object.

There are inevitably installation errors that must be considered in the process of building the system. Hence, the camera may have a certain offset relative to the object in the x, y, and z directions at the initial moment. Specifically, in the actual case, the initial coordinates of the camera are C $(x_0, y_0, h_0)$, and $x_0$, $y_0$, and $h_0$ are unknown. In this case, the coordinate motion relationship among the axes can be modified as follows:

$$
\begin{aligned}
\tan(\alpha) &= \frac{\Delta x - x_0}{h_0 + z_0 - \Delta z} \\
\tan(\beta) &= \frac{\Delta y - y_0}{\sqrt{(\Delta x - x_0)^2 + (h_0 + z_0 - \Delta z)^2}} \\
\Delta z &= h_0 + z_0 - \sqrt{(h_0 + z_0)^2 - (\Delta x - x_0)^2 - (\Delta y - y_0)^2}
\end{aligned}
\tag{2}
$$

To realize the calculation of $x_0$, $y_0$, and $h_0$, we manually adjust the camera and the object to calibrate the parameters $x_0$, $y_0$, and $h_0$. Specifically, the *x*-, *y*-, *z*-axes move randomly in multiple positions, then the rotation axis is adjusted artificially to ensure that the object is in the center of the camera field of view. After repeating many times, the mapping relationship between multiple groups of $x_0$, $y_0$, $h_0$ and $\Delta x$, $\Delta y$, $\Delta z$, $\alpha$, $\beta$ is obtained, as shown in Table 1. Since $\Delta x$, $\Delta y$, $\Delta z$, $\alpha$, $\beta$ are all known parameters,

a number of constraint equations about $x_0$, $y_0$, $h_0$ can be established by introducing Equation (2), and the initial parameters $x_0$, $y_0$, and $h_0$ can be calculated by solving the equations.

**Table 1.** Acquisition points of each axis.

| ID | $\Delta x$ (mm) | $\Delta y$ (mm) | $\Delta z$ (mm) | $\alpha$ (°) | $\beta$ (°) |
|---|---|---|---|---|---|
| 1 | −32.005 | 40.737 | −62.9 | −9.71 | 13.1875 |
| 2 | −48.803 | 59.681 | −51.652 | −15.5475 | 19.575 |
| 3 | −67.044 | 82.066 | −36.371 | −22.9025 | 27 |
| 4 | −79.439 | 89.984 | −18.315 | −29.5975 | 30.75 |
| 5 | −85.581 | 34.003 | −41.329 | −27.7375 | 11.4425 |
| 6 | −21.941 | −27.195 | −62.9 | −6.895 | −8.2525 |
| 7 | −41.033 | −48.84 | −49.987 | −13.725 | −15.425 |
| 8 | −60.088 | −43.401 | −44.696 | −20.0725 | −13.625 |
| 9 | −75.295 | −59.2 | −27.972 | −27.0875 | −19.1775 |
| 10 | −84.397 | −77.108 | −13.542 | −32.455 | −25.36 |
| 11 | 45.658 | 22.681 | −62.9 | 14.055 | 7.01 |
| 12 | 63.27 | 46.472 | −51.282 | 20.26 | 14.7425 |
| 13 | 82.88 | 62.9 | −33.966 | 28.1025 | 20.3125 |
| 14 | 90.946 | 75.48 | −24.975 | 31.85 | 24.4275 |
| 15 | −49.765 | −67.451 | −39.849 | −17.61 | −21.5625 |
| 16 | −72.113 | −41.736 | −36.371 | −24.8475 | −13.34 |
| 17 | −86.062 | −30.599 | −31.783 | −29.58 | −9.5825 |
| 18 | −67.895 | −48.396 | −35.039 | −23.82 | −15.65 |
| 19 | 99.271 | −67.044 | −12.58 | 35.41 | −22.86 |
| 20 | 77.552 | −75.036 | −25.16 | 26.8125 | −24.83 |

$z_0 = 62.9$ mm when the camera is in focus.

By solving several constraint equations and averaging the calculated results, the solution is $x_0 = 0.5241$ mm, $y_0 = 0.1761$ mm, $h_0 = 121.4259$ mm.

To verify the effect of the modified equation, the above-calculated $x_0$, $y_0$, $h_0$ were substituted into Equation (2), then 15 groups of test points were collected. $\Delta z$, $\alpha$, and $\beta$ were calculated according to Equations (1) and (2) and compared with the actual measured value; the results are shown in Figure 6. We evaluated the accuracy of the calculated value by error. The smaller the error, the closer the calculated value was to the measured value, and the more obvious the effect of the equation's correction. It can be seen from the figure that the error values after correction were less than those before correction, indicating that the error correction had a certain effect. However, the calculated $\Delta z$, $\alpha$, and $\beta$ still had some deviation from the measured values.

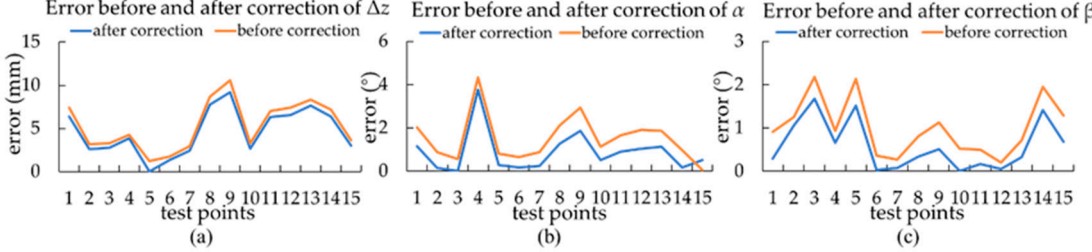

**Figure 6.** Error before and after correction: (**a**) error before and after correction of $\Delta z$; (**b**) error before and after correction of $\alpha$; (**c**) error before and after correction of $\beta$.

### 3.1.2. Camera Position Calculating Method by Data Fitting

The relationship between each axis needs to be determined by data fitting. The reasons are as follows: On the one hand, the mathematical modeling method is usually assumed to be carried out under ideal conditions. However, there will be camera installation errors in the actual measurement

process. On the other hand, there is also the error caused by the rotation axes when calculating the camera rotation angle through a simple geometric relationship. The specific process is as follows: Controlling the *x*-, *y*-, and *z*-axes to move $\Delta x$, $\Delta y$, and $\Delta z$, then control the rotation axes *a* and *b* to rotate the corresponding $\alpha$ and $\beta$ angles respectively to ensure that the camera optical axis points (the record the values of $\Delta x$, $\Delta y$, $\Delta z$, $\alpha$, $\beta$); Repeating the above steps and recording the values of multiple groups of $\Delta x$, $\Delta y$, $\Delta z$, $\alpha$, $\beta$. Finally, fitting the corresponding relationship between each axis. The experimental acquisition points are shown in Table 1. The polynomial fitting of degrees 1, 2, 3, and 4 were performed, and the fitting error is shown in Table 2.

**Table 2.** Polynomial fitting error.

| Degrees | $\Delta z$ (mm) | $\alpha$ (°) | $\beta$ (°) |
|---|---|---|---|
| 1 | 15.9265 | 1.36364 | 0.355234 |
| 2 | 2.0463 | 1.231859 | 0.372423 |
| 3 | 1.7221 | 0.280749 | 0.189076 |
| 4 | 1.5537 | 0.257831 | 0.177617 |

It can be seen from Table 2 that the error decreased with the increase of fitting times. When the fitting degrees were 4, the errors of $\Delta z$, $\alpha$, and $\beta$ reached the minimum, but the errors were only slightly less than the errors of degrees 3. Considering the computational complexity and overfitting phenomenon, the fitting degrees were finally selected as 3. Finally, the relationship between the axes was obtained by data fitting as shown in Equation (3).

$$
\begin{cases}
\alpha = 8.127 \times 10^{-8} \times \Delta x^3 - 3.562 \times 10^{-8} \times \Delta y^3 - 4.424 \times 10^{-7} \times \Delta x^2 - 2.074 \times 10^{-6} \times \Delta y^2 + 1.864 \times 10^{-8} \times \Delta x^2 \times \Delta y + 9.825 \times 10^{-8} \times \\
\qquad \Delta x \times \Delta y^2 - 2.286 \times 10^{-6} \times \Delta x \times \Delta y + 0.0051 \times \Delta x + 0.0001905 \times \Delta y + 0.001021 \\
\beta = -6.336 \times 10^{-9} \times \Delta x^3 + 2.108 \times 10^{-8} \times \Delta y^3 - 4.218 \times 10^{-7} \times \Delta x^2 - 6.109 \times 10^{-10} \times \Delta y^2 + 3.101 \times 10^{-8} \times \Delta x^2 \times \Delta y - 8.65 \times 10^{-10} \\
\qquad \times \Delta x \times \Delta y^2 - 5.084 \times 10^{-7} \times \Delta x \times \Delta y - 2.631 \times 10^{-5} \times \Delta x + 0.005444 \times \Delta y + 0.002839 \\
\Delta z = -2.708 \times 10^{-7} \times \Delta x^3 + 1.319 \times 10^{-5} \times \Delta y^3 + 0.004024 \times \Delta x^2 + 0.003225 \times \Delta y^2 - 5.464 \times 10^{-6} \times \Delta x^2 \times \Delta y - 2.844 \times 10^{-6} \times \Delta x \\
\qquad \times \Delta y^2 + 0.0002364 \times \Delta x \times \Delta y - 0.004346 \times \Delta x - 0.1057 \times \Delta y - 69.67
\end{cases}
\tag{3}
$$

### 3.1.3. Comparison of the Two Methods

To find the optimal solution between these two methods, the calculation results of the two methods were compared. The method of obtaining $\Delta x$, $\Delta y$, $\Delta z$, $\alpha$, and $\beta$ values was the same as the method in Section 3.1.2. The values of $\Delta z$, $\alpha$, and $\beta$ were calculated by introducing $\Delta x$ and $\Delta y$ into Equations (2) and (3), respectively, and the calculated values were compared with the actual values.

It can be seen from Figure 7 that the errors of $\Delta z$, $\alpha$, and $\beta$ of method 2 were less than those of method 1, indicating that method 2 is more suitable for the calculation of camera pose. Therefore, the system uses the result of data fitting (Equation (3)) to calculate the camera pose.

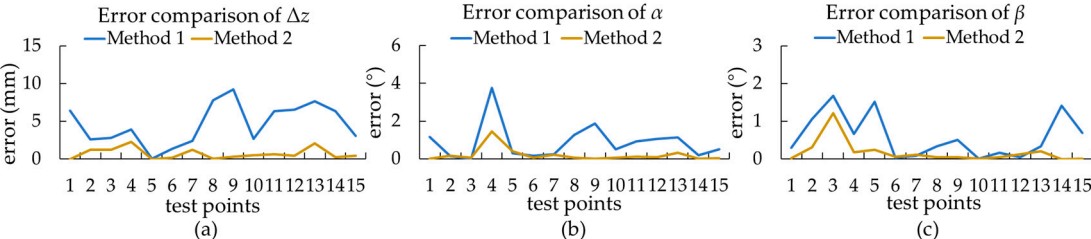

**Figure 7.** Error comparison of the two methods: (**a**) Error comparison of $\Delta z$; (**b**) error comparison of $\alpha$; (**c**) error comparison of $\beta$; method 1 is the camera position calculating method by mathematical modeling, method 2 is the camera position calculating method by data fitting.

### 3.2. Random Acquisition Path Optimization

The random distribution of acquisition points in the system space is the premise of simulating actual shooting situations. More specifically, the common shooting angle is generally in a certain range with the top of the object as the center point, and the probability of the occurrence of large-angle tilt is relatively small. Therefore, the random points on the $x$- and $y$-axes were set to obey normal distribution, and then the coordinates of $z$-, $a$-, $b$-axes were obtained according to the above calculation method of camera space pose. At this time, the coordinate points of the five axes of $x$, $y$, $z$, $a$, and $b$ constitute the random acquisition points that obey normal distribution in the system space, as shown in Figure 8.

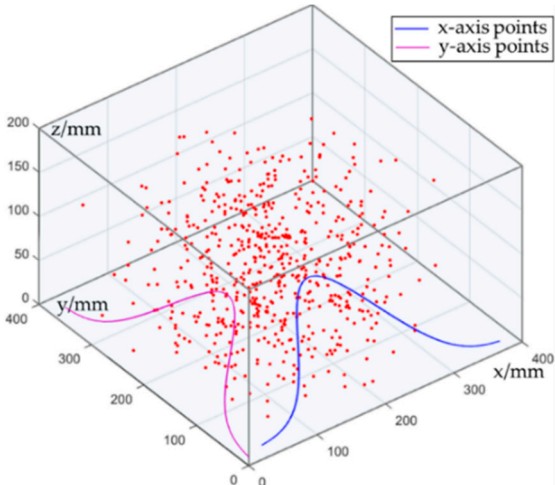

**Figure 8.** Random sampling points with the normal distribution.

After generating random acquisition points, a "detour" will occur when the system collects images in the order of generating random points, the acquisition time will be consumed. Therefore, it is necessary to optimize the path of the generated random acquisition points, so that the system can complete image acquisition in a short time. According to the actual acquisition requirements, the camera is required to start from the origin (initial position), then pass through all acquisition points only once. Eventually, it takes the least time to return to the original acquisition point, which is in line with the travelling salesman problem (TSP) proposed by Dantzig et al. [27]. The algorithm has two path optimization methods: local optimization and global optimization as shown in Figure 9. The advantages and disadvantages of these two algorithms are shown in Table 3. Although the path obtained by the global optimal search method is optimal, its algorithm takes a long time and has high computational complexity, which is not suitable for the case of hundreds or even thousands of points. Therefore, the system adopts the local optimal search method to optimize the acquisition path.

**Table 3.** Comparison of local optimal algorithm and global optimal algorithm.

| Index | Local Optimal | Global Optimal |
|---|---|---|
| Time-consuming | Less | More |
| Optimal path | No | Yes |
| Algorithm complexity | $n$ | $n!$ |

Comparative experiments before and after path optimization are carried out to evaluate the impact of path optimization algorithm on system acquisition time. The number of acquisition points is set as 5, 10, 15, . . . , 100 (spacing is 5). Recording the acquisition time of the system before and after the path optimization of the acquisition algorithm. Each point is tested three times to get the average value. The final acquisition time is shown in Figure 10. The search principle of the above TSP path

optimization algorithm is the shortest distance between two points. However, when the system uses the optimal path search, the distance between two points in space is not the shortest path of the axis movement. The reason is that under the condition of the same speed of each axis, when the point with the largest distance among the five coordinate points moves, the remaining coordinate points will also complete the movement. Hence, we take the maximum distance max $(x, y, z, a, b)$ of five coordinate points as the search principle.

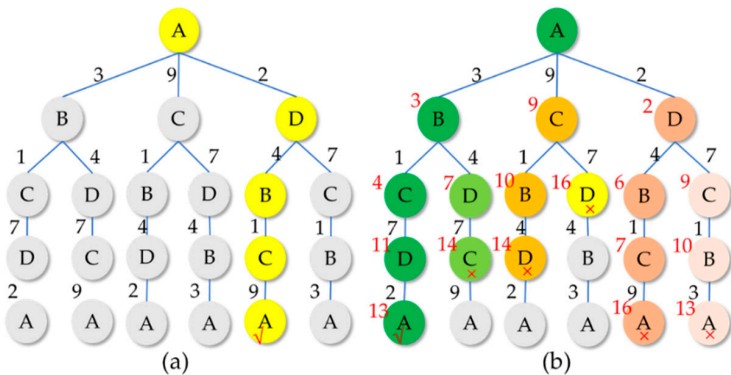

**Figure 9.** The travelling salesman problem (TSP) algorithm [27]: (**a**) local optimal, (**b**) global optimal.

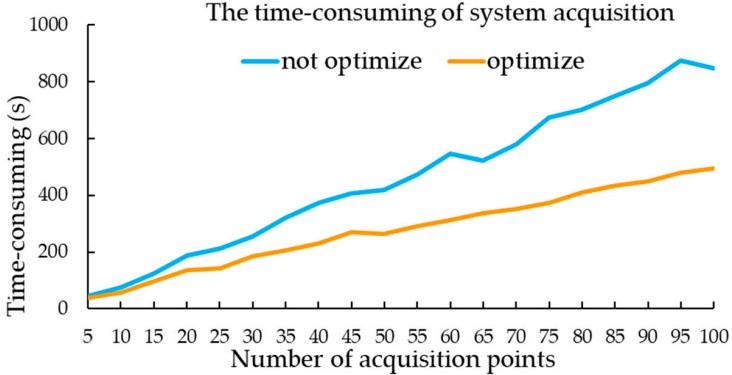

**Figure 10.** System acquisition time before and after path optimization.

As can be seen from Figure 10, the system acquisition time is increasing with the increase in the number of acquisition points. The path acquisition time before optimization increases significantly. However, the path acquisition time after optimization increases slowly. Meanwhile, it can be seen from Figure 10 that the time consumption of path acquisition before the optimization is about 1.6 times of that after optimization. In conclusion, it is very necessary to optimize the acquisition path when the number of acquisition points is large, the path optimization algorithm adopted in this paper had an obvious effect.

## 4. Experiments and Discussions

Since the first industrial revolution, the textile industry has become the most fundamental industry and is closely related to people's lives. With the development of textile technology, the application field of fabric is also expanding. Fabric has penetrated all aspects of industrial production and life such as military, medical, architecture, aerospace, etc. At present, there are tens of thousands of known fabrics on the market. It is of guiding significance for production and sales control to determine a category of fabric quickly and accurately in the process of fabric production and sales. However, due to the variety of fabrics, it is very difficult to search by manual or traditional digital image processing technology. In addition, a fabric itself is a periodic structure based on pattern design, which is more

suitable for image recognition and classification by deep learning methods. In this work, the fabric is taken as the experimental object, and the fabric data is collected by the multi-DOF automatic image acquisition system, and the fabric classification is realized by the deep learning method.

### 4.1. Construction of Multi-DOF Automatic Image Acquisition System

The multi-DOF automatic image acquisition system is shown in Figure 11. Figure 11a shows a hardware structure of the multi-DOF automatic image acquisition system, which mainly includes motion control and light source control. The motion control component mainly includes an electric guide rail, driver and 8-axes motion control card. The light source control component mainly includes flat light source, ring light source, light source controller and STM32 single chip microcomputer. The key components and related parameters used in the system are shown in Table 4.

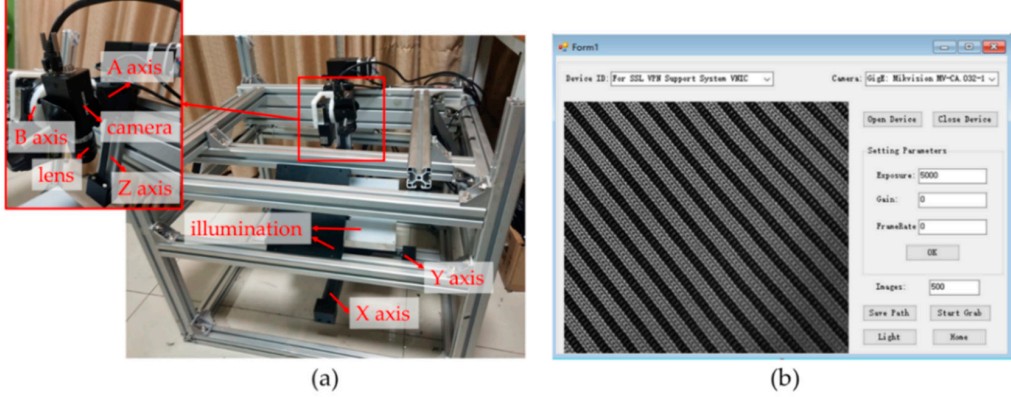

**Figure 11.** The multi-DOF automatic image acquisition system: (**a**) Hardware structure of the multi-DOF automatic image acquisition system; (**b**) The UI of the multi-DOF automatic image acquisition system.

**Table 4.** Key components and related parameters of the system.

|  | Key Components | Model | Parameters |
|---|---|---|---|
| Motion control | Electric guide rail | JD45P | Travel: 300 mm |
|  | Driver | DM442 | Two-phase stepping motor driver |
|  | 8 axes motion control card | IMC408E | 8 axes |
| Illumination control | Flat light source | HF-FX160 | DC12V, white light |
|  | Ring light source, | YC-DR6836WL | DC12V, white light |
|  | Light source Controller | CCS PD-3012-8 | Input AC100-240V, output: DC12V, power: 25 W |
|  | Single chip Microcomputer | STM32F103ZET6 | Cotex-M3 core chip |

Figure 11b shows the user interface designed according to the multi-DOF automatic image acquisition system. The interface mainly includes an image display area and parameter setting area. The image display area is used to display the fabric image in real-time. The parameter setting area mainly includes camera parameters (exposure, frame rate, gain, etc.), illumination (light source, brightness), image saving path, and the number of collected images.

### 4.2. Image Acquisition Experiment

To verify the representativeness of images collected by this system, this section divides the image acquisition experiment into two parts, which are image acquisition by using this system and image acquisition by manual. The camera used in the experiment was a Hikvision industrial area array camera, the model was MV-CA032-10GM, the resolution was $1920 \times 1440$, the focal length of the lens was 25 mm, and 30 classes of fabrics were collected. The size of each fabric was 15 cm × 30 cm, and the field of view of the camera was 3.5 cm × 2.6 cm.

### 4.2.1. Image Acquisition by the System

Before collecting the fabric image, it was necessary to calibrate the speed of $x$, $y$, $z$, $a$, $b$ axes of the system. Additionally, the fabric surface should be kept as flat as possible in the acquisition process, so as to obtain high-definition images. Five hundred images were collected for each class of fabric. Part of the image collected by the system is shown in Figure 12.

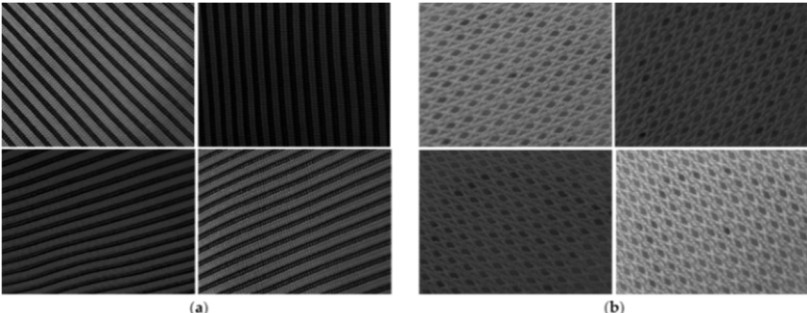

**Figure 12.** Part of the data collected by the system: (**a**) Fabric 4; (**b**) Fabric 26.

### 4.2.2. Collect Images Manually

To obtain the contrast dataset, under the ideal conditions (the industrial camera was fixed vertically at a certain distance above the fabric, and the ambient light was uniform when collecting the image), the image was collected by manually controlling the camera. Part of the image collected by the system is shown in Figure 13. Fifty images were collected for each class of fabric. Images were expanded to 500 through image processing to keep consistent with the number of images collected by the system. Since the image enhancement technology based on deep learning [28] will increase the calculation time to a certain extent, and the quality of the generated image is difficult to control, it is difficult to meet the needs of automatic training. Therefore, this paper uses the traditional data augmentation technology to expand the collected data. Typical traditional data augmentation techniques include flipping, clipping, rotation, translation, scaling, histogram equalization, enhancing contrast or brightness, random erasure, etc., [18–21]. In order to ensure that the augmented data are consistent with the real data as much as possible, combined with the actual environmental conditions in the fabric detection process, this paper finally used rotation, stretching, brightness transformation, and other methods to augment the data.

Table 5 shows the collection methods and collection time-consuming of the three types of datasets in image acquisition experiment. The experimental results show that the system takes about 9 min to collect 500 images (a kind of fabric) and about 3 min to collect 50 images (a kind of fabric) manually. On average, the image acquisition speed of the system is faster than that of manual acquisition obviously, and the angle and illumination of the manual acquisition are single. The image shooting angle and illumination obtained by the system are variable, and the image richness is higher.

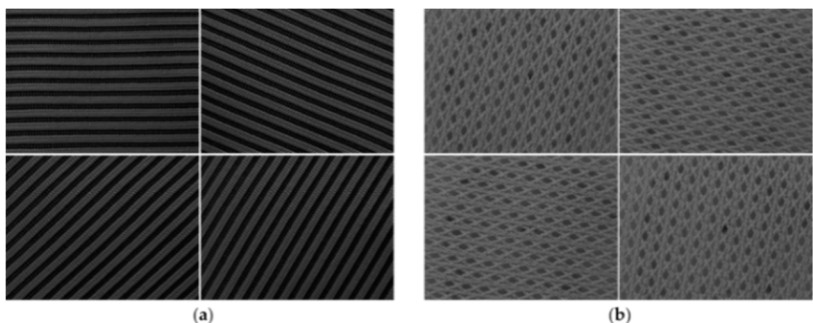

**Figure 13.** Part of data collected manually: (**a**) Fabric 4; (**b**) Fabric 26.

**Table 5.** Acquisition methods and time-consuming image data.

| Datasets | Acquisition Method | Size | Time-Consuming | |
|---|---|---|---|---|
| | | | One Class | One Image |
| dataset1 | Manual acquisition | $50 \times 30$ | 3 min | 3.6 s |
| dataset2 | Dataset1, data augmentation | $500 \times 30$ | - | - |
| dataset3 | System acquisition | $500 \times 30$ | 9 min | 1.1 s |

### 4.3. Datasets Comparison Experiment

#### 4.3.1. Experiment Description

To further verify whether the validity and representative of the obtained datasets, this section shows results of comparative experiments on the datasets obtained by the above methods. The image classification network model used was ResNet [27] in the experiment, which successfully solved the problem of gradient explosion or gradient disappearance with the deepening of network depth by introducing residual block. Due to the fact that the amount of data was not too large ($500 \times 30$), we chose a shallow network structure resnet18 which also has good stability. The specific experimental process is shown in Figure 14. The above three different image datasets dataset1, dataset2, and dataset3 are scaled to adapt to different resolutions, then separately put these three types of training datasets into the same classification network model for training. Under the same training strategy (the number of iterations is 20, the learning rate was 0.001, the batch size was 32, and the image size was $224 \times 224$), the same image classification network (ResNet18) learned the distribution state of dataset1, dataset2, and dataset3 separately, then obtained three different models: model1, model2, and model3. Specifically, model 1 was trained from a small number of data collected manually (dataset1). Model 2 was trained from the augmentation data (dataset2), and model 3 was trained from the dataset collected by the system (dataset3). Finally, the performance of the three models were tested by using the test datasets, and the conclusions were drawn through the comparative analysis of the test results.

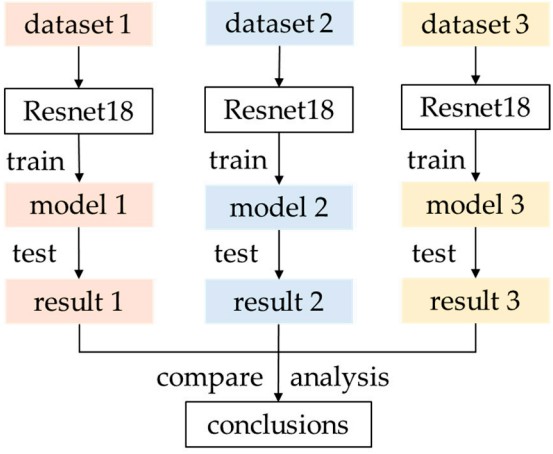

**Figure 14.** The flow of datasets comparison experiment.

#### 4.3.2. Test Datasets Composition

In the practical application process, the image acquisition method was arbitrary in order to verify that the data collected by the system were representative enough and the trained model could identify the data collected under various conditions. Firstly, the test dataset should contain an ideal collection environment, where the camera is facing the fabric and the illumination is uniform, and it is named as the manual test dataset. Secondly, the test dataset should contain complex collection scenes (collection environment built by the system) with complex lighting environments and arbitrary shooting angles,

and it is named as the system test dataset. Additionally, the shooting equipment will be different in actual applications. For example, people are more accustomed to taking pictures with mobile phones in real life. Therefore, the test dataset should include images taken by mobile phone, which is named as mobile phone test dataset. Some images of the above three test datasets are shown in Figure 15.

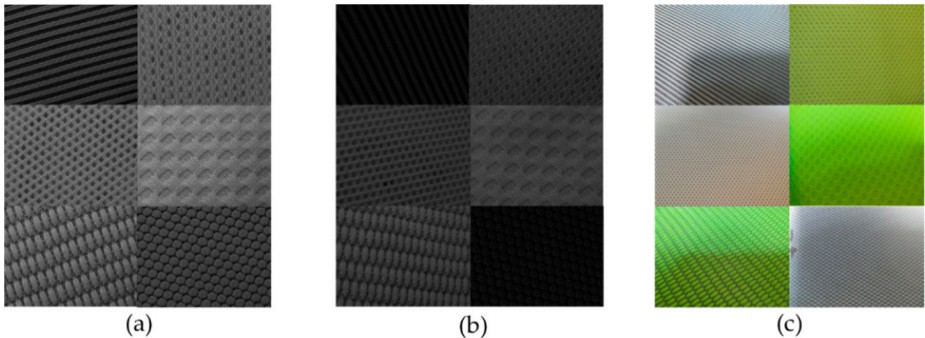

**Figure 15.** Part of the test dataset images: (**a**) manual test dataset; (**b**) system test dataset; (**c**) mobile phone test dataset.

### 4.3.3. Result Analysis

Figure 16 shows the recognition effect of model 1, model 2, and model 3 on the dataset collected manually, the dataset collected by the system, and the dataset collected by the mobile phone. Each row in the table represents the recognition effect of a model for different test dataset, and each column represents the recognition effect of different models for the same test dataset. They are represented by a confusion matrix, where the abscissa represents the predicted label of the model for the test data, and the ordinate represents the true label of the test data. When the number of predicted labels equal to true labels, the more data on the diagonal, the better the recognition effect of the model. As can be seen from Figure 16, the data distribution of model 1 in the three test datasets is very scattered, indicating that the performance of model 1 is poor. In model 2, although the data distribution has been improved, there are still some data outside the diagonal in the system test dataset and mobile phone test dataset, which shows that the recognition performance of model 2 for the system and mobile phone test dataset is not high, that is, when the shooting environment is complex, it is difficult for model 2 to detect the data. From the confusion matrix of model 3 for the three test datasets, it can be seen that the data distribution of the three types of test datasets were all concentrated on the diagonal, which shows that model 3 has good recognition effect on the manual, system, and mobile test datasets.

As can be seen from the Figure 17, the recognition accuracy of model 1, which was trained with fewer data (dataset 1), was generally lower than 60%. This shows the performance of the model obtained through the small sample training data was poor. For the recognition rate of model 2, which was trained with the augmentation of the data (dataset 2), the accuracy improved for the data obtained under ideal conditions. However, the recognition accuracy rate for the test dataset collected by the system and mobile phone was still less than 80%. This shows that the dataset augmented through the data augmentation technology was quite different from the dataset collected in actual applications. Only relying on the data augmentation technology to augment the data, the trained model cannot be applied to actual industrial production. The recognition accuracy of model 3, which was trained with the data obtained by this system (dataset 3), was over 91% for the three types of test datasets. In particular, the accuracy of the test dataset collected by the system and mobile phone was significantly improved, indicating that the dataset collected by the system contained most of the collection scenarios in the application. The model trained from the dataset collected by the system could recognize images taken in most complex environments.

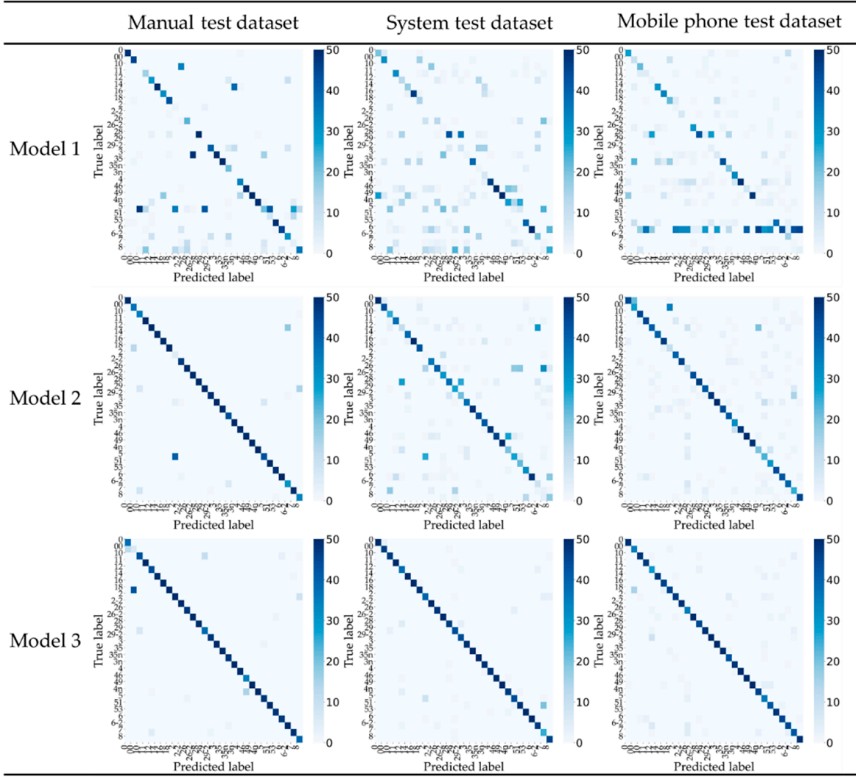

**Figure 16.** Confusion matrixes of different test datasets on model 1, model 2, and model 3.

The results showed that this system can not only obtain a large amount of data in a short time but also include most of the actual data collection. The model trained by the system acquisition data has a better recognition effect for the images obtained by common acquisition methods in practical application and has a good recognition effect for the data collected in complex scenes. At the same time, it also shows that the model trained by our system can be embedded into mobile terminals, such as mobile apps, for image recognition.

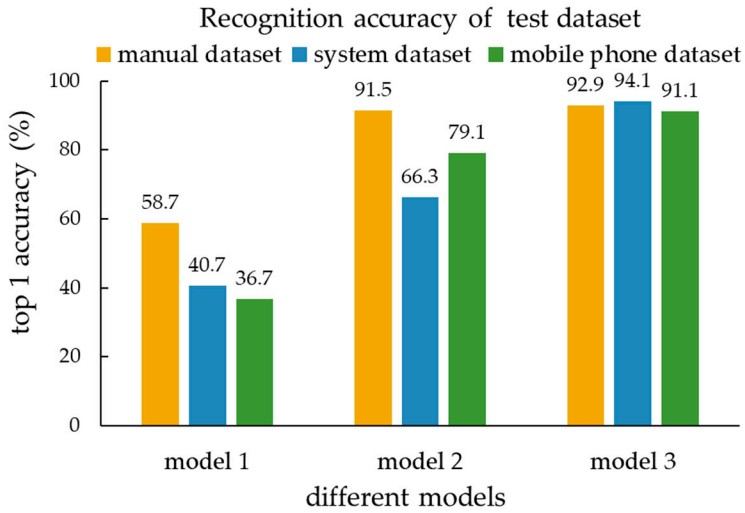

**Figure 17.** Test results of three models.

The method proposed in this paper forms a mature acquisition system in the actual project (Figure 18a) and has been applied to a large number of industrial field tests. As many as dozens of

light sources were built in the system to simulate a more complex acquisition environment. It was mainly used for defect image acquisition of complex structural parts such as automobile combination teeth (Figure 18b). In addition, the proposed method can also be applied to image acquisition of high reflective workpieces (Figure 19a) and transparent workpieces (Figure 19b). In the specific application scenario, the actual acquisition system can be set-up according to our method.

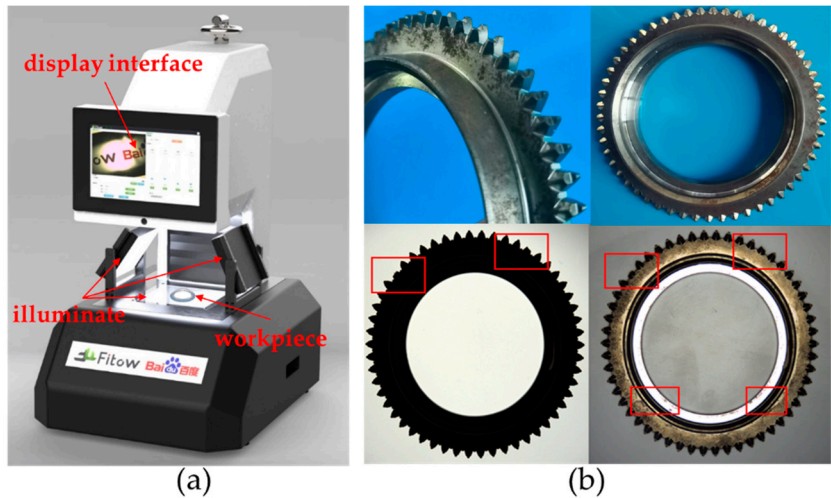

**Figure 18.** Specific system application: (**a**) Mature acquisition system; (**b**) automobile combination teeth.

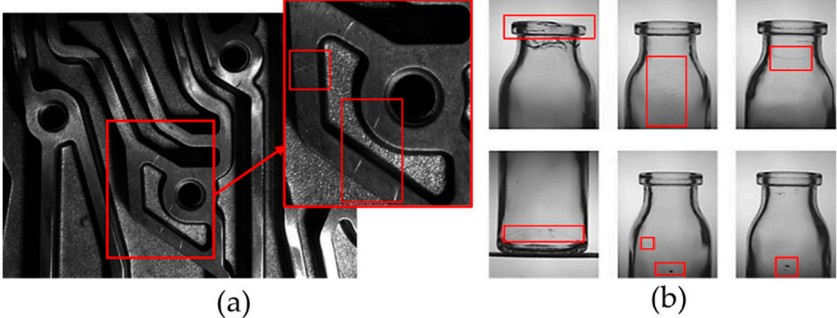

**Figure 19.** Other application objects: (**a**) highly reflective workpiece; (**b**) transparent workpiece.

## 5. Conclusions

Aiming at the problem that datasets are scarce in the process of deep learning in industrial applications, this paper proposed a data augmentation method based on multi-freedom automatic image acquisition and built a multi-degree of freedom image automatic acquisition system for deep learning. The main conclusions are summarized as follows:

(1) A multi-degree of freedom automatic image acquisition system for deep learning was built to simulate the actual image acquisition situation. In this system, the multi-directional light source was arranged for random lighting, and the multi-degree of freedom motion axis was designed to carry out random motion of the object;

(2) In the process of image acquisition, rich and diverse data can be obtained in a short time; this work calculated the camera position and optimized the random acquisition path. The system can collect 500 images (a class of fabric) in only 9 min;

(3) A deep learning model was used to verify the type of obtained datasets by different methods. The results showed that the recognition accuracy of images collected by the system for different scenes was more than 91%. The construction of the system further promotes the application of deep learning in industrial production.

**Author Contributions:** X.Z. was the leader of this research. He proposed the basic idea and participated in the discussion. N.Y. participated in the research discussion and experimental design. L.C. developed the algorithm and conducted the experiments and wrote and revised the manuscript. H.Y. participated in algorithm development and modification. Z.Z. help to set-up an acquisition model and organize manuscripts. X.Y. and L.Z. were responsible for building the system. All authors have read and agreed to the published version of the manuscript.

**Funding:** This research was funded by the National Key Research and Development Program of China (Grant No. 2017YFA0701200), Science Challenge Program (Grant No. TZ2018006-0203-01), and Tianjin Natural Science Foundation of China (Grant No. 19JCZDJC39100). Thanks to FITOW (Tianjin) Detection Technology Co. LTD. and Baidu Co., Ltd. for their trust and support for the proposed method and application of this method in actual industrial use.

**Conflicts of Interest:** The authors declare no conflict of interest.

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
