# Peer review of "A Data Augmentation Method for Deep Learning Based on Multi-Degree of Freedom (DOF) Automatic Image Acquisition"

_applsci, doi:10.3390/app10217755_

Round 1
Reviewer 1 Report
The paper presents a multi-degree of freedom automatic image acquisition system to support building of large size image databases needed for appropriate DNN training. The concept is quite original, my main concern however is a lack of practical application of the proposed system. It enables acquisition of large number of optical images that represent objects of limited size (by the way, limitations regarding visualized object size are not defined). Thus, the system will be useful if analysed object only if:
- is a solid, portable and physically fits to the system;
- can be visualised using visible electromagnetic spectrum.
Such system is perfect to acquire images of artificial textures (as shown in the paper), but this is not very convincing case. Thus, the Authors should provide more realistic examples of system application and convince the readers that proposed solution is really useful in practice.
Reviewer 2 Report
The paper presents a system for collecting of objects (surfaces) images for different camera positions (in relation to the object) and for different lighting conditions. For collecting of images, a specialized system was prepared, enabling the positioning of the camera in five degrees of freedom in changing lighting conditions. The paper presents the results of "deep learning" with the use of fabric images recorded on a specialized system and with the use of a mobile phone. In Conclusion, it was stated that the proposed system implements a "process of image acquisition" and thus "rich and diverse datasets can be obtained in a short time" ("the system can collect 500 images ... in only 9 minutes"). In addition, the paper describes how to determine the camera position and optimize the camera movement path.
The main subject of the paper is unclear. If we consider "the multi-DOF automatic image acquisition system" as its central part, then, in the opinion of the reviewer, there is no "classic" research on the quality of positioning. The term "error" appears in the paper (eg. on Figs. 6 and 7, Table2) - but there is no definition of this term. On the other hand, if the method of image acquisition control, including acquisition path optimization, is the subject of the paper, the presented research results are modest. Also, the verification of the representativeness of images collected by the proposed system is limited.
Despite the doubts expressed, the paper is an interesting example of obtaining data necessary for the effective use of "deep learning".
Reviewer 3 Report
Dear authors, good to see your paper, It is an interesting idea however there are some concerns for this reviewer which i believe needs to be addressed. Comments can be seen in the attached file.

Reviewer 4 Report
The manuscript "A multi-degree of freedom (DOF) automatic image 2 acquisition system for deep learning" shows new and interesting results and it appears as a good work. The article is well-organized and easy to understand. The authors have well synthesized the literature in the introduction and the topic is original and of great interest. The authors have created a multi-DOF automatic image acquisition system which can solve the problem of image dataset acquisition in industrial production, but also in most research and multidisciplinary fields, seens it can allows a faster acquisition of a greater number of images collected at different angles and in different lighting conditions.
Author Response
Thank you very much for your review and positive comments.
Round 2
Reviewer 1 Report
I am grateful for Authors' answer, however, it does not dispel my doubts. I still don't know what real applications of the proposed system would be. To be useful in industry, it should carefully reflect all specific conditions and parameters of production line. For example, one of issue important issues, not discussed in this work, is illumination system and their properties. It influences much the quality of obtained photos and may differ significantly for different industrial applications. It is not clear how the fixed illumination system proposed in the paper would fit to the real ones.
Such system would be of interest in students’ labs to build big databases of images to train deep networks. However, since the name of the journal to which the authors apply is the word "Applied", it means that the submitted works must have some practical application. The Authors did not provide any such example - they probably don't know any - thus confirming the key limitation of the proposed system.
Reviewer 3 Report
The authors have paid much attention to the comments and tried to address the concerns. Thus, this reviewer has no more concerns except to recommend the reviewers to study the methods proposed in the literature to overcome the limitations of limited availability of training data/samples/examples to train conventional or deep models, such as interactive learning, active learning, self-learning, etc. Good luck.
Author Response
Thanks for your affirmation positive comments of our research. We will further verify our proposed method in specific application scenarios based on your comments and improve the system. Thank you again.
Round 3
Reviewer 1 Report
Dear Authors,
if you performed any analyses you mention in amended part of the paper (regarding gearwheels and glass bottels), please describe them in more details and show the results in similar way as it was done for fabric. This way you will convince me that your system has some practical applications.